# Plastome Evolution, Phylogenomics, and DNA Barcoding Investigation of *Gastrochilus* (Aeridinae, Orchidaceae), with a Focus on the Systematic Position of *Haraella retrocalla*

**DOI:** 10.3390/ijms25158500

**Published:** 2024-08-04

**Authors:** Peng Zhou, Wan-Shun Lei, Ying-Kang Shi, Yi-Zhen Liu, Yan Luo, Ji-Hong Li, Xiao-Guo Xiang

**Affiliations:** 1Key Laboratory of Poyang Lake Environment and Resource Utilization Ministry of Education, School of Life Sciences, Nanchang University, Nanchang 330031, China; zhoupengrjcck@163.com (P.Z.); 405600220103@email.ncu.edu.cn (W.-S.L.); 5604120013@email.ncu.edu.cn (Y.-K.S.); liuyizhen@ncu.edu.cn (Y.-Z.L.); 2Xishuangbanna Tropical Botanical Garden, Chinese Academy of Sciences, Mengla 666300, China; luoyan@xtbg.org.cn; 3Kadoorie Farm and Botanic Garden, Hong Kong Lam Kam Road, Tai Po, New Territories, Hong Kong 999077, China; jli@kfbg.org

**Keywords:** *Gastrochilus*, *Haraella retrocalla*, molecular markers, phylogenomics, plastome evolution

## Abstract

*Gastrochilus* is an orchid genus containing about 70 species in tropical and subtropical Asia with high morphological diversity. The phylogenetic relationships among this genus have not been fully resolved, and the plastome evolution has not been investigated either. In this study, five plastomes of *Gastrochilus* were newly reported, and sixteen plastomes of *Gastrochilus* were used to conduct comparative and phylogenetic analyses. Our results showed that the *Gastrochilus* plastomes ranged from 146,183 to 148,666 bp, with a GC content of 36.7–36.9%. There were 120 genes annotated, consisting of 74 protein-coding genes, 38 tRNA genes, and 8 rRNA genes. No contraction and expansion of IR borders, gene rearrangements, or inversions were detected. Additionally, the repeat sequences and codon usage bias of *Gastrochilus* plastomes were highly conserved. Twenty hypervariable regions were selected as potential DNA barcodes. The phylogenetic relationships within *Gastrochilus* were well resolved based on the whole plastome, especially among main clades. Furthermore, both molecular and morphological data strongly supported *Haraella retrocalla* as a member of *Gastrochilus* (*G. retrocallus*).

## 1. Introduction

*Gastrochilus* D. Don (Aeridinae, Vandeae, Epidendroideae, Orchidaceae) is an epiphytic orchid genus comprising about 70 species, and widely distributed in tropical and subtropical Asia [1,2,3,4], with a species diversity center in the South-East Asian archipelago [5,6]. Because of its high morphological diversity and brightly colored flowers, it has potential horticultural value for pot culture, hanging baskets, and tree mounting [2,7]. Additionally, it can be used as a medicine to treat mastitis, body aches, and detoxification due to its phytochemical production such as bioactive alkaloids [2,8].

Recently, phylogenetic analyses confirmed that the genus *Gastrochilus* was monophyletic, but the infrageneric relationships were not completely resolved. Based on ITS and four plastid markers (*atpI-atpH*, *matK*, *psbA-trnH*, and *trnL-F*), Zou et al. [9] showed that the nine *Gastrochilus* species formed a clade with high Bayesian inference supporting value, and similar results were obtained in other phylogenetic studies with ITS and several chloroplast DNA markers. Unfortunately, the relationships within *Gastrochilus* were not consistent with each other [3,4,6]. Specifically, Liu et al. [6] proposed that *Gastrochilus* can be divided into five clades based on five DNA regions (ITS, *matK*, *psbA-trnH*, *psbM-trnD*, and *trnL-F*). Recently, Zhang et al. [4] also employed the same five markers and divided *Gastrochilus* into six sections. However, these infrageneric classifications of this genus were not supported by Li et al. [3] based on the combination of ITS and seven chloroplast DNA markers. Further, Liu et al. [10] reconstructed the phylogenetic relationships within the *Cleisostoma-Gastrochilus* clades (Aeridinae) based on 68 plastid genes, and strongly supported the idea that *Gastrochilus* is close to *Pomatocalpa*. Particularly, the position of *Gastrochilus retrocallus* (Hayata) Hayata (*Haraella retrocalla* (Hayata) Kudo) was controversial. *G. retrocallus* is an epiphytic species endemic to Taiwan, China. It was firstly described as *Saccolabium retrocallum* by Hayata [11], and then recognized as a member of *Gastrochilus* as *G. retrocallus* [12]. Because this species lacks a saccate hypochile, Kudo [13] established a new genus, *Haraella*, including two species *H. retrocalla* and *H. odorata* Kudo. Later, Smith [14] transferred *H. odorata* into *Gastrochilus* as *G. odoratus* (Kudo) J.J.Sm. After a detailed morphological examination, Tsi [1] revised the genus *Gastrochilus* and treated *G. odoratus* and *G. retrocallus* as synonyms of *H. retrocalla*. However, this taxonomic treatment has not been adopted by some taxonomists, and they still recognized them as *G. retrocallus* (e.g., [2,15], https://powo.science.kew.org/, accessed on 7 June 2024). Additionally, recent phylogenetic studies based on combined ITS and plastid DNA markers have still not effectively addressed its systematic position. For example, Zou et al. [9] indicated that *H. retrocalla* was sister to *Gastrochilus* with low supporting values (PP_BI_ = 0.65, BS_ML_ < 50), while Liu et al. [6] deemed it nested in *Pomatocalpa* (PP_BI_ = 1.00). Therefore, it is crucial to employ more effective molecular markers to resolve the phylogenetic relationships within *Gastrochilus*, and further clarify the taxonomic position of *H. retrocalla*.

Plastomes have recently played crucial roles in investigating plant phylogenetic relationships (e.g., [16,17,18]). For example, phylogenetic analyses indicated that whole plastomes realized a fast and efficient identification of the relationships within *Polygonatum* (Asparagaceae) [19]. In particular, plastomes were successful in elucidating intergeneric relationships within Orchidaceae. Phylogenetic analysis based on the complete plastome indicated that *Aerides* (Aeridinae) was monophyletic and can be divided into three major clades [20]. Moreover, robust phylogenetic relationships of *Angraecum* (Angraecinae) were established based on plastomes [21]. Additionally, plastomes are advantageous in resolving the phylogenetic relationships in other Orchidaceae genera, such as *Pholidota* (Coelogyninae) [22] and *Epidendrum* (Epidendrinae) [23].

In this study, we newly sequenced, assembled, and annotated the plastomes of five *Gastrochilus* species, and combined them with 11 previously reported plastomes of *Gastrochilus* to conduct comparative analyses, DNA barcoding investigation, and phylogenetic reconstruction within *Gastrochilus*. Our objectives were (1) to investigate general features and understand the evolutionary pattern of *Gastrochilus* plastomes; (2) to identify some hypervariable regions as potential DNA barcodes for future species identification within *Gastrochilus*; and (3) to explore the phylogenetic relationships of *Gastrochilus* and discuss the systematic position of *H. retrocalla*.

## 2. Results

### 2.1. Plastome Features and Gene Contents

The plastomes of the five *Gastrochilus* species displayed a typical quadripartite structure (Figure 1). The plastome sizes of *Gastrochilus* ranged from 146,183 bp (*G. fuscopunctatus* (Hayata) Hayata) to 148,666 bp (*G. acinacifolius* Z.H.Tsi) (Appendix A). The plastomes contained a pair of inverted repeats (IRs; 25,725–26,025 bp), a large single-copy (LSC; 83,125–85,759 bp) region, and a small single-copy (SSC; 10,805–11,331 bp) region. The GC content of the plastomes ranged from 36.7% to 36.9%. The GC content of the IR and LSC regions ranged from 43.0% to 43.2% and 33.9% to 34.2%, respectively, while the GC content of the SSC region ranged from 28.0% to 28.6% (Appendix A). Each *Gastrochilus* plastome consisted of 120 genes, comprising 74 protein-coding genes (CDSs), 38 transfer RNA (tRNA) genes, and 8 ribosomal RNA (rRNA) genes.

### 2.2. Plastome Structural Variations

The IR boundary map was generated by comparing the plastomes of 16 *Gastrochilus* species using IRscope (Figure 2). At the junction between LSC and IRb (JLB), the *rpl22* gene in all species spanned from LSC to IRb with 31–44 bp departed in IRb. Moreover, the *trnN* and *rpl32* genes of *Gastrochilus* plastomes were found adjacent to the junction between the SSC and IRb (JSB), while none of them spanned the junction. In addition, the *ycf1* gene spanned from SSC to IRa in 15 plastomes at the junction between SSC and IRa (JSA), with a range of 11 to 195 bp, while the *ycf1* gene of *G. guangtungensis* Z.H.Tsi was entirely located in the SSC region. As for the junction between IRa and LSC (JLA), the *rps19* and *psbA* genes were detected on the left and right side of the JLA line, respectively. The collinearity analysis revealed no gene rearrangements or inversions in the *Gastrochilus* plastomes (Appendix A).

### 2.3. Examination of Repeats and Codon Usage Bias

The results of SSRs, tandem repeats, and dispersed repeats in the 16 *Gastrochilus* plastomes are shown in Figure 3 and Appendix A. A total of 763 SSRs were detected in the 16 *Gastrochilus* plastomes, of which 480 SSRs (62.91%) were located in the LSC region, 183 SSRs (23.98%) in the SSC region, and 100 SSRs (13.11%) in the two IR regions. Six types of SSRs (mono-, di-, tri-, tetra-, penta-, and hexa-nucleotide repeats) were identified, with 541 SSRs (70.90%) being mono-nucleotide type, particularly A and T repeat motifs. We also identified 98 di-nucleotide repeats (12.84%), 52 tri-nucleotide repeats (6.82%), 53 tetra-nucleotide repeats (6.95%), 15 penta-nucleotide repeats (1.97%), and 4 hexa-nucleotide repeats (0.52%). Mono-, di-, tri-, and tetra-nucleotide repeat categories were observed in all species, while penta-nucleotide repeats were absent in 5 species. Hexa-nucleotide repeats were only present in *G. japonicus* (Makino) Schltr. and *G. prionophyllus* H.Jiang, D.P.Ye & Q.Liu. In each species, a total of 40 (*G. sinensis* Z.H.Tsi) to 58 (*G. retrocallus*) SSRs were identified.

A total of 706 tandem repeats were detected, ranging from 30 (*G. japonicus*) to 53 (*G. retrocallus*). There were 626 long repeats in the *Gastrochilus* plastomes, comprising four types: palindrome, forward, reverse, and complement. Among them, palindrome repeats were the dominant type of long repeats, followed by forward repeats, with percentages of 65.81% and 30.19%, respectively, while reverse and complement repeats only accounted for only 3.51% and 0.48%, respectively. All types of long repeats were detected only within two species (*G. distichus* (Lindl.) Kuntze and *G. prionophyllus*). The total number of long repeats ranged from 31 (*G. japonicus*) to 51 (*G. retrocallus*) in each *Gastrochilus* plastome.

To quantify the degree of the codon usage bias, we estimated the relative synonymous codon usage (RSCU) ratio of 16 *Gastrochilus* plastomes using CodonW, which is visualized in Appendix A. There were 30 preferred codons (RSCU > 1), 2 non-preferred codons (RSCU = 1), and 32 less frequently used codons (RSCU < 1). Most of the preferred codons typically ended with A or U, except for UUG. Moreover, leucine (Leu, encoded by UUA, UUG, CUU, CUC, CUA, and CUG) was the most frequently encoded amino acid, while cysteine (Cys, encoded by UGU and UGC) had the lowest frequency. The codons AGA and UUA exhibited the highest RSCU values, with average values of 1.92 and 1.89, respectively, while the codons CGC and CGG had the lowest RSCU values, with average values of 0.31 and 0.34, respectively.

### 2.4. Plastome Sequence Divergence and Barcoding Investigation

The divergence of the complete plastome sequences among the 16 *Gastrochilus* species was analyzed using the mVISTA with *Pomatocalpa spicatum* Breda as reference (Appendix A). The whole genome alignment revealed that sequence variations in the conserved non-coding regions (CNS; colored in pink bars) were greater than that in the protein-coding regions (exon; colored in purple bars). The variation rates of both coding regions and non-coding regions in the two IR regions were lower than those in the LSC and SSC regions. Additionally, the non-coding intergenic regions were highly divergent, such as *trnS^GCU^-trnG^GCC^*, *rpl32-trnL^UAG^*, and *psaC-rps15*, while the rRNA genes were highly conserved compared with other genes.

To further explore the mutation hotspots of *Gastrochilus* plastomes to develop specific DNA barcodes, nucleotide diversity (Pi) values were calculated using DnaSP6 (Figure 4). The average Pi value among the 16 plastomes was 0.00719, with the IR region averaging 0.00206, the LSC region averaging 0.00804, and the SSC region averaging 0.01388. According to the ranking of Pi values, the top ten hypervariable regions of whole plastomes were identified: *rpl32-trnL^UAG^*, *ccsA-ndhD*, *matK-rps16*, *trnS^GCU^-trnG^GCC^*, *psaC-rps15*, *rbcL-accD*, *accD-psaI*, *rps16-trnQ^UUG^*, *trnE^UUC^-trnT^GGU^*, and *petA-psbJ*. In terms of the 68 CDSs, we also found the top ten hypervariable regions: *ycf1*, *rpoC2*, *ccsA*, *matK*, *rpoA*, *accD*, *rpl20*, *rps16*, *rps11*, and *rps8* (ranked by Pi values). These hypervariable regions may be used as DNA barcodes for further phylogenetic analyses and species identification.

### 2.5. Phylogenomic Analysis

The topologies based on the whole plastome (excluding IRa) and 68 CDSs were basically concordant. BI and ML analyses also yielded nearly identical topologies, with some differences in the supporting values of certain nodes (Figure 5A,B). The species of *Gastrochilus* formed a well-supported monophyletic group (PP_BI_ = 1.00, BS_ML_ = 100), which was revealed as a sister to *Pomatocalpa*. The *G. retrocallus* (formerly treated as *Haraella retrocalla*) diverged firstly as clade I, and the remaining species of *Gastrochilus* could be divided into three monophyletic clades with strong supporting values (PP_BI_ = 1.00, BS_ML_ = 100). Specifically, *G. gongshanensis* Z.H.Tsi and *G. obliquus* (Lindl.) Kuntze formed clade II. Clade III consisted of a pair of sister groups with strong supporting values (PP_BI_ = 1.00, BS_ML_ = 100): one included *G. formosanus* (Hayata) Hayata, *G. sinensis*, and *G. distichus*, and the other included *G. acinacifolius* and *G. guangtungensis*. Finally, the remaining species formed clade IV, which also included two monophyletic subclades with strong supporting values (PP_BI_ = 1.00, BS_ML_ = 100). However, clade II and III formed as sister groups with weaker support based on the two datasets.

In addition, to test the resolution of potential DNA barcodes for phylogenetic analyses, we reconstructed the phylogenetic relationships of *Gastrochilus* based on the top 10 hypervariable regions in the whole plastome and 68 CDSs. The two phylogenetic trees presented the same topologies (Figure 5C,D). All sampled *Gastrochilus* species formed a monophyletic group with high support values (PP_BI_ = 0.98 and 1, BS_ML_ = 85 and 98, respectively). Remarkably, the monophyly of clades II, III, and IV was strongly supported (PP_BI_ = 1.00, BS_ML_ = 100), while the relationships among the four clades were poorly resolved.

## 3. Discussion

### 3.1. Plastome Evolution within Gastrochilus

In this study, we firstly reported five *Gastrochilus* plastomes and provided genetic resources for understanding the evolution of plastomes in this group. All *Gastrochilus* plastomes had a typical quadripartite structure (Figure 1), consisting of one LSC region, one SSC region, and two IR regions, which were similar to the other orchids and most of the angiosperms (e.g., [21,24]). Limited variation in plastome size was detected among *Gastrochilus* species: *G. fuscopunctatus* possessed the smallest plastome at 146,183 bp, and *G. acinacifolius* had the largest at 148,666 bp. Plastome size falls within the previously reported range of Orchidaceae plastomes, which ranged from 19,047 bp (*Epipogium roseum* (D.Don) Lindl.) [25] to 212,688 bp (*Cypripedium tibeticum* King ex Rolfe) [26], and near to the size of other genera in Aeridinae, such as *Aerides* (147,244–148,391 bp) [20], *Paraphalaenopsis* (147,311–149,240 bp) [27], and *Chiloschista* (143,223–145,463 bp) [28]. Similar GC content and gene number were found among *Gastrochilus* plastomes in this study, which are similar with other Aeridinae species (e.g., [20,27,28]). In addition, our study showed that all *ndh* genes in *Gastrochilus* plastomes were lost or pseudogenic, which has been observed universally in Epidendroideae [29], such as *Aerides* [20], *Angraecum* [21], and *Bulbophyllum* [30].

No visible gene rearrangement was detected among *Gastrochilus* plastomes (Appendix A), which was also observed in other orchid genera (e.g., *Aerides* [20]; *Epidendrum* [23]). In addition, our results revealed that all IR boundaries were conserved without distinct contraction or expansion (Figure 2). We only observed a slight difference in the JSA boundary regions of *Gastrochilus* plastomes. In most *Gastrochilus* plastomes, the *ycf1* gene extended from SSC to IRa for 11–195 bp except *G. guangtungensis*. Similarly, the *ycf1* gene in the SSC region of other orchids (such as *Epidendrum* [23] and *Pholidota* [22]) was also observed crossing over JSA, extending into the IRa region. Our results also indicated that codon usage bias was highly conserved among 16 *Gastrochilus* plastomes (Appendix A), which was consistent with previous studies of codon preference in Orchidaceae (e.g., [21,23,30]), and further demonstrates the high level of plastome conservation in *Gastrochilus*.

In addition, our results visually showed that species with closer phylogenetic relationships tend to have more similar plastome structures and sequence divergence patterns (Figure 2 and Appendix A). For example, the whole *ycf1* gene of *G. guangtungensis* was located in SSC, and *ycf1* in its sister species only extended from SSC to IRa for 11 bp, while the three species with the longest span (195 bp) of *ycf1* formed a subclade (Figure 2). Additionally, unique sequence divergence at about 99.5 kb and 132 kb only appeared in clade II, and the species with significant sequence divergence at about 54 kb and 60.5 kb formed as a monophyly (Appendix A). Similar phenomena were also observed in other studies, such as *Chiloschista* [28], *Pholidota* [22], and *Epidendrum* [23]. Therefore, we speculated that the structure and sequence divergence of plastomes may also contain important evolutionary information.

### 3.2. Genetic Molecular Markers

SSRs were often used as genetic molecular markers in phylogenetic studies of closely related species (e.g., [31,32,33]). In this study, a total of 763 SSRs were identified in the plastomes of 16 *Gastrochilus* species, with 70.90% of them being mono-nucleotide repeats. A/T SSRs were found to be more abundant compared to G/C SSRs (Figure 3; Appendix A), which may result from a bias towards A/T in plastomes [30]. Most di- to hexa-nucleotide SSRs among *Gastrochilus* species were specific to each species (Figure 3). These SSRs were widely distributed through the plastome, and more than half of SSRs (62.91%) were located in the LSC region, which is similar to other angiosperm plastomes (e.g., [24,34]). The diversity and richness of SSR types vary across different species and may be attributed to the genetic variations among species [30].

The variation in the non-coding region was higher than that in the coding region, with the variation in the SC region being higher than that in the IR region (Figure 4A and Appendix A), which is consistent with other angiosperm lineages (e.g., [24,34]). Top ten hypervariable regions of whole plastome (*rpl32-trnL^UAG^*, *ccsA-ndhD*, *matK-rps16*, *trnS^GCU^-trnG^GCC^*, *psaC-rps15*, *rbcL-accD*, *accD-psaI*, *rps16-trnQ^UUG^*, *trnE^UUC^-trnT^GGU^*, and *petA-psbJ*) and CDSs (*ycf1*, *rpoC2*, *ccsA*, *matK*, *rpoA*, *accD*, *rpl20*, *rps16*, *rps11*, and *rps8*) were identified in *Gastrochilus* (Figure 4). In addition, most of the 20 hypervariable regions were also identified in other orchid lineages with high nucleotide diversity values, such as *rpl32-trnL^UAG^*, *rbcL-accD*, *ycf1*, *and matK* in *Bulbophyllum* [30]; *trnS^GCU^-trnG^GCC^*, *rbcL-accD*, *ycf1*, *rps16*, and *rps8* in *Angraecum* [21]; *rpl32-trnL^UAG^*, *trnS^GCU^-trnG^GCC^*, *matK-rps16*, and *rps16-trnQ^UUG^* in Coelogyninae [35]; and *psaC-rps15*, *accD-psaI*, *trnE^UUC^-trnT^GGU^*, *ycf1*, *ccsA*, and *matK* in *Chiloschista* [28]. Interestingly, the mVISTA percent identity plot showed some intergenic regions which were highly divergent, such as *trnS^GCU^-trnG^GCC^*, *rpl32-trnL^UAG^*, and *psaC-rps15*. These hypervariable regions were also identified in the nucleotide diversity analysis with high Pi values.

Hypervariable regions explored for phylogenetic and identification analyses have been reported in Orchidaceae [36,37], and many orchid lineages such as *Cleisostoma-Gastrochilus* clades [10] and *Chiloschista* [28]. In this study, the phylogenetic relationships solved based on ten hypervariable regions of the whole plastome and CDSs were nearly the same as those based on the whole plastome and 68 CDSs (Figure 5); therefore, we propose that the top ten hypervariable regions of the whole plastome and CDSs might be powerful markers for the phylogenetic analysis of *Gastrochilus*.

### 3.3. The Systematic Position of Haraella retrocalla and Phylogenomics of Gastrochilus

The phylogenetic results based on the whole plastome (excluding IRa), 68 CDSs, and ten hypervariable regions strongly supported that *H. retrocalla* (*G. retrocallus*) was grouped with *Gastrochilus* species with high supporting values (Figure 5). Especially the phylogenetic result of the whole plastome supported well the idea that *H. retrocalla* was a sister to *Gastrochilus*. *H. retrocalla* was previously recognized as a member of *Haraella* [1,5,13], or *Gastrochilus* [2,12,15]. In this study, 14 morphological characters (representing stem, leaf, and flower) of 42 *Gastrochilus* species, *H. retrocalla*, and three *Pomatocalpa* species were analyzed (Appendix A). Except for the absence of saccate hypochile in *Haraella*, *H. retrocalla* and *Gastrochilus* species are very similar in stem, leaf, and flower size, as well as in the number of flowers in one inflorescence. Additionally, principal component analysis (PCA) also revealed that *H. retrocalla* has no differentiation from *Gastrochilus* species, but is obviously distinct from *Pomatocalpa* (Appendix A). Therefore, our results supported the inclusion of *H. retrocalla* into *Gastrochilus* as *G. retrocallus* based on morphological and molecular evidence.

All four phylogenetic results of *Gastrochilus* strongly supported the monophyly of *Gastrochilus* species (Figure 5). *G. retrocallus* was sister to other *Gastrochilus* species with high supporting values based on the whole plastome. The phylogenetic relationships of *Gastrochilus* were better resolved based on the whole plastome than other datasets. In addition, the monophyly of the other three clades were fully supported (PP_BI_ = 1.00, BS_ML_ = 100%) based on the whole plastome and 68 CDSs, which was consistent with the results in Liu et al. [10] based on 68 CDSs. Moreover, the relationships between the three clades were not completely resolved, which may be due to the rapid divergence of *Gastrochilus* during the late Miocene [3]. Many studies indicated that the phylogenetic relationships of recent radiation plant lineages, such as *Rhododendron* [38], *Astragalus* [39], and *Acacia*, were not clear [40]. Therefore, we speculated that more samplings and more molecular data (such as mitogenome and transcriptomes) are needed to better understand the phylogenetic relationships within *Gastrochilus*.

## 4. Materials and Methods

### 4.1. Sampling and Sequencing

In this study, five new plastomes of *Gastrochilus* species were obtained, including *G. acinacifolius*, *G. distichus*, *G. malipoensis* X.H.Jin & S.C.Chen, *G. prionophyllus*, and *G. yunnanensis* Schltr. Another eleven published plastomes of *Gastrochilus* were downloaded from GenBank and the annotations in Geneious v9.1.4 [41] were manually updated. Additionally, two species of the *Gastrochilus* clade in Aeridinae (*Pomatocalpa spicatum* and *Trichoglottis philippinensis* Lindl.) were selected as outgroups based on Liu et al. [10]. The detailed information of the samples is listed in Appendix A.

Leaf samples of five new sampled *Gastrochilus* species were cultivated and obtained from the Xishuangbanna Tropical Botanical Garden, Chinese Academy of Sciences, Yunnan. We extracted total DNA from silica gel-dried leaves using the modified CTAB method [42]. Library construction was performed with the NEB Next^®^ Ultra DNA Library Prep Kit (NEB, Ipswich, MA, USA), and libraries for paired-end 150 bp sequencing were created using an Illumina HiSeq 2000 platform at the Kunming Institute of Botany, Chinese Academy of Sciences (Yunnan, China). Finally, approximately 4 Gb of data were obtained for each species.

### 4.2. Plastome Assembly and Annotation

We assessed the quality of raw sequence reads in FastQC v0.11.9 [43] and filtered the adapters and low-quality reads using Trimmomatic v0.39 [44]. Then, the clean reads were assembled using GetOrganelle v1.7.3.2 [45], and the assembled genomes were checked and visualized in Bandage v0.7.1 [46]. Finally, we obtained five high-quality and complete plastomes.

The obtained plastomes were annotated and manually checked in Geneious v9.05 [41] with *G. calceolaris* (Buch.-Ham. ex Sm.) D.Don (NC_042686) and *Amborella trichopoda* Baill. (NC_005086) as references. The annotation circle maps were drawn using OGDRAW (https://chlorobox.mpimp-golm.mpg.de/OGDraw.html, accessed on 7 June 2024). The assembled and annotated chloroplast genome information (GenBank accession numbers: PP963516-PP963520) was uploaded to the NCBI database.

### 4.3. Structure and Sequence Divergence Analyses

To evaluate the possible expansion and contraction of the IR boundary, the genes on the boundary regions of LSC/IRb/SSC/IRa were visualized using IRscope v3.1 [47]. Moreover, to detect the gene arrangement, Mauve v1.1.3 [48] plugin in Geneious v9.1.4 [41] was used to conduct the collinearity analysis with default parameters. The online program mVISTA [49] was used to analyze the sequence divergence of *Gastrochilus* plastomes using the *Pomatocalpa spicatum* (MN124411) plastome as a reference.

### 4.4. Repetitive Sequence and Codon Usage Analyses

Three types of repeat sequences of sixteen plastomes of *Gastrochilus* were analyzed, including SSRs, tandem repeats, and long repeats. Specifically, SSRs were detected using MISA v2.1 [50] and visualized using R packages “ggpubr v0.6.0” [51] and “ggplot2 v3.4.3” [52]. Different lengths of SSRs were determined by a settled minimum threshold of 10, 5, 4, 3, 3, and 3 repeat units for mono-, di-, tri-, tetra-, penta-, and hexa-nucleotides, respectively. Tandem repeats were found using Tandem Repeats Finder v0.9 [53]. Finally, we identified the four types of long repeats (including forward, reverse, complement, and palindromic) in REPuter [54] following the detailed parameter settings in Cauz-Santos et al. [55].

Additionally, we estimated the RSCU ratio of sixteen *Gastrochilus* plastomes using CodonW v1.4.2 [56]. RSCU > 1 indicates a positive codon usage bias, while RSCU < 1 indicates a less frequent usage [57]. Finally, R package “pheatmap” [58] was used to create the heatmap for the RSCU analysis.

### 4.5. Evolutionary Hotspots and Phylogenetic Analyses

In order to avoid the impact of two IR regions on phylogenetic reconstruction, we identified the hypervariable regions and conducted the phylogenetic analyses with IRa excluded. We identified the hypervariable regions of 16 *Gastrochilus* species based on the following two matrices: (1) whole plastomes with IRa excluded and (2) 68 CDSs. The two matrices were aligned using MAFFT v7 [59] and manually adjusted in BioEdit v7.0 [60]. We further evaluated the Pi value using DnaSP v6.12.03 [61] with sliding window analysis by setting step size to 200 bp and window length to 800 bp.

After adding two species as outgroups, the phylogenetic relationships were inferred based on following four matrices: (1) whole plastome sequences (excluding IRa); (2) concatenation of 68 CDSs; (3) top ten hypervariable regions of whole plastomes (excluding IRa); (4) top ten hypervariable regions of 68 CDSs. The four matrices with 18 species were aligned by MAFFT v 7 [59] and manually adjusted in BioEdit v7.0 [60]. Phylogenetic analyses were carried out using maximum likelihood (ML) and Bayesian inference (BI) methods for each combined matrix, respectively. We conducted ML analyses in RAxML v8.2.12 [62], and the best-fit model of sequence evolution was estimated using the Akaike information criterion in jModeltest v2.1.4 [63]. BI analyses were performed in MrBayes v3.2.6 [64] with 10,000,000 generations and sampled every 1000 generations. The majority rule (>50%) consensus tree was obtained after removing the first 25% of the sampled trees as “burn-in”.

### 4.6. Morphological Character Analysis

To test whether morphological differentiation corroborates the phylogenetic relationship of *H. retrocalla*, we collected 14 morphological traits of stem, leaves, and flowers that were considered taxonomically important in systematic studies of *Gastrochilus*. All morphological traits were collected from existing studies (e.g., [1,5,65,66]). Then, we conduct PCA in R package “vegan” [67] to delimitate genus boundaries, included the morphological data from *H. retrocalla*, 42 species of *Gastrochilus*, and 3 species of *Pomatocalpa*, which is the sister group of *Gastrochilus* [10]. In this analysis, the first two coordinates were selected to draw the PCA scatter plot. The 14 morphological traits of 46 species are provided in Appendix A.

## 5. Conclusions

In this study, we obtained the plastomes of five *Gastrochilus* species (*G. acinacifolius*, *G. distichus*, *G. malipoensis*, *G. prionophyllus*, and *G. yunnanensis*) and compared them with another eleven *Gastrochilus* plastomes to investigate plastome evolution and phylogenetic relationships. The plastome characteristics and comparative analysis results indicated that the genomic size, GC content, gene content, IR boundary, structure, repeat sequences, and codon usage bias of *Gastrochilus* plastomes are highly conserved. Additionally, according to the ranking of Pi values, the top ten hypervariable regions of whole plastome and 68 CDSs were identified for DNA barcodes in *Gastrochilus*, including 10 non-coding regions (*rpl32-trnL^UAG^*, *ccsA-ndhD*, *matK-rps16*, *trnS^GCU^-trnG^GCC^*, *psaC-rps15*, *rbcL-accD*, *accD-psaI*, *rps16-trnQ^UUG^*, *trnE^UUC^-trnT^GGU^*, and *petA-psbJ*) and 10 CDSs (*ycf1*, *rpoC2*, *ccsA*, *matK*, *rpoA*, *accD*, *rpl20*, *rps16*, *rps11*, and *rps8*), respectively. Combined with the morphological data, our results strongly supported the idea that *Haraella retrocalla* was included in *Gastrochilus* (*G. retrocallus*). Based on the whole plastome (excluding IRa), *G. retrocallus* was the basal clade and the remaining 15 *Gastrochilus* species can be divided into three monophyletic clades with high supporting values.

## Figures and Tables

**Figure 1 ijms-25-08500-f001:**
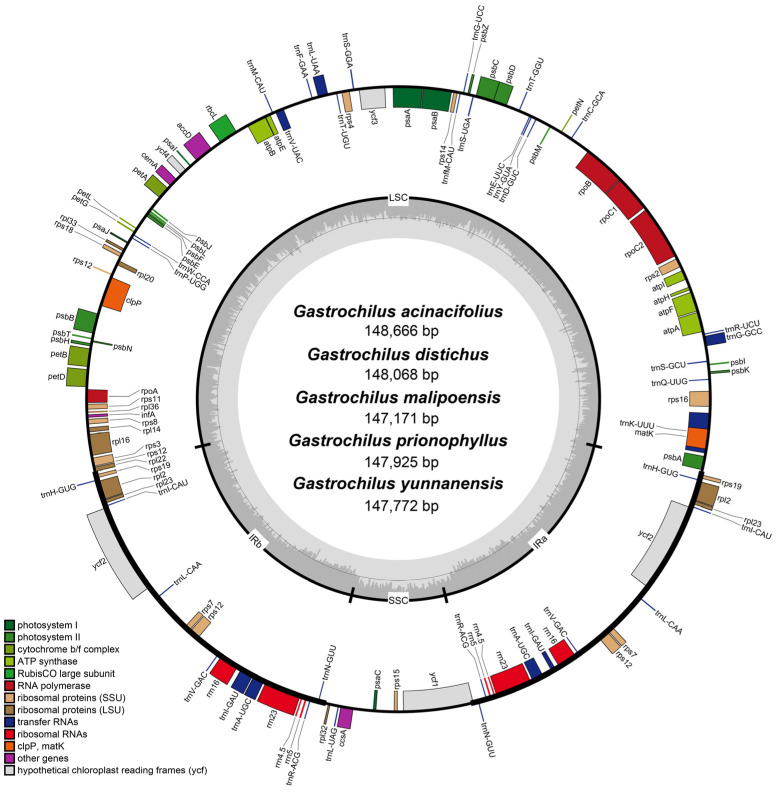
Plastome structure of five *Gastrochilus* species. The darker gray in the inner circle corresponds to the GC content. Bars of different colors indicate different functional groups. Genes on the inside of the circle are transcribed clockwise, while genes annotated outside the circle are transcribed counterclockwise. LSC: large single-copy region; SSC: small single-copy region; IRa and IRb: two inverted repeat regions.

**Figure 2 ijms-25-08500-f002:**
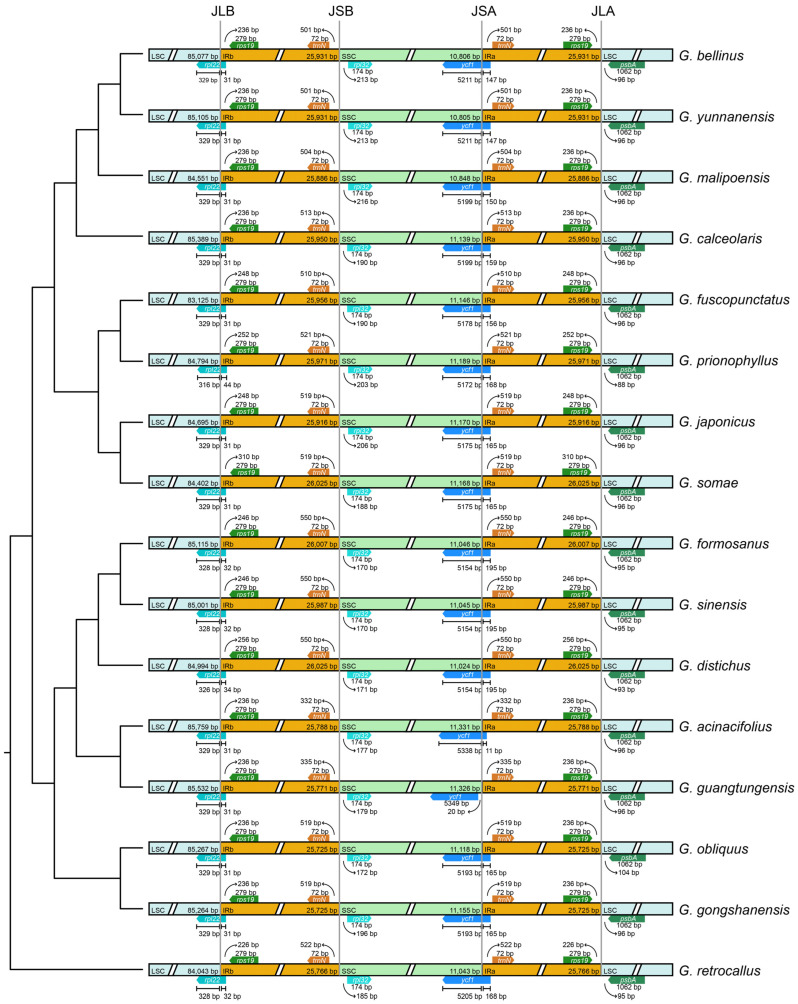
Comparison of the boundaries between the LSC, SSC, and IR regions in the sixteen *Gastrochilus* plastomes. The topology on the left was the ML tree based on plastomes (excluding IRa). JLB: LSC/IRb junctions; JSB: SSC/IRb junctions; JSA: SSC/IRa junctions; JLA: LSC/IRa junctions.

**Figure 3 ijms-25-08500-f003:**
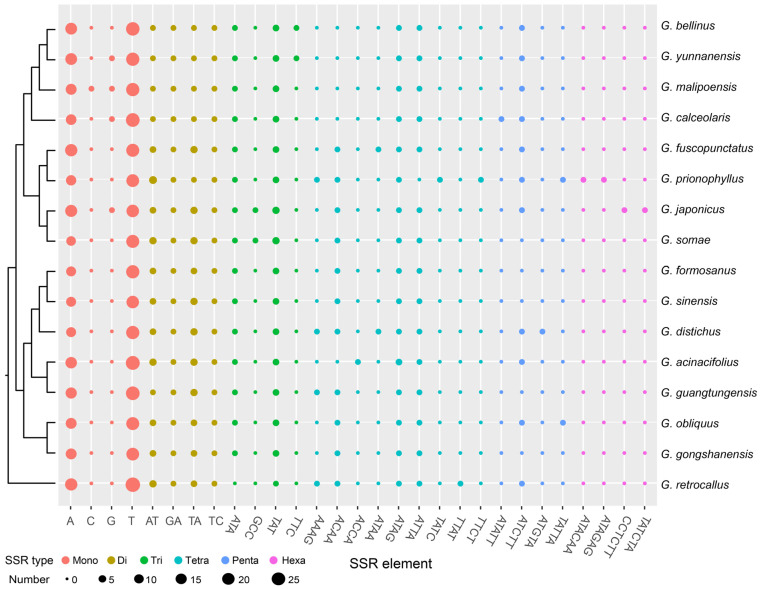
Plot of each SSR repeat pattern number of 16 *Gastrochilus* plastomes. The topology on the left is the ML tree based on plastomes (excluding IRa).

**Figure 4 ijms-25-08500-f004:**
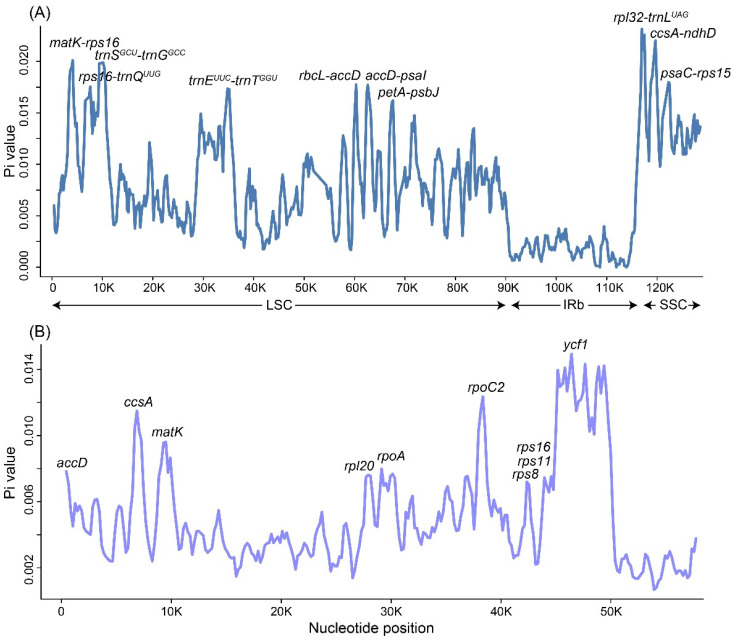
Sliding window analysis of nucleotide diversity for *Gastrochilus* plastomes. (**A**) The nucleotide diversity of the whole plastome (excluding IRa). (**B**) The nucleotide diversity of 68 protein coding sequences. Top ten hypervariable regions of the two datasets were annotated respectively.

**Figure 5 ijms-25-08500-f005:**
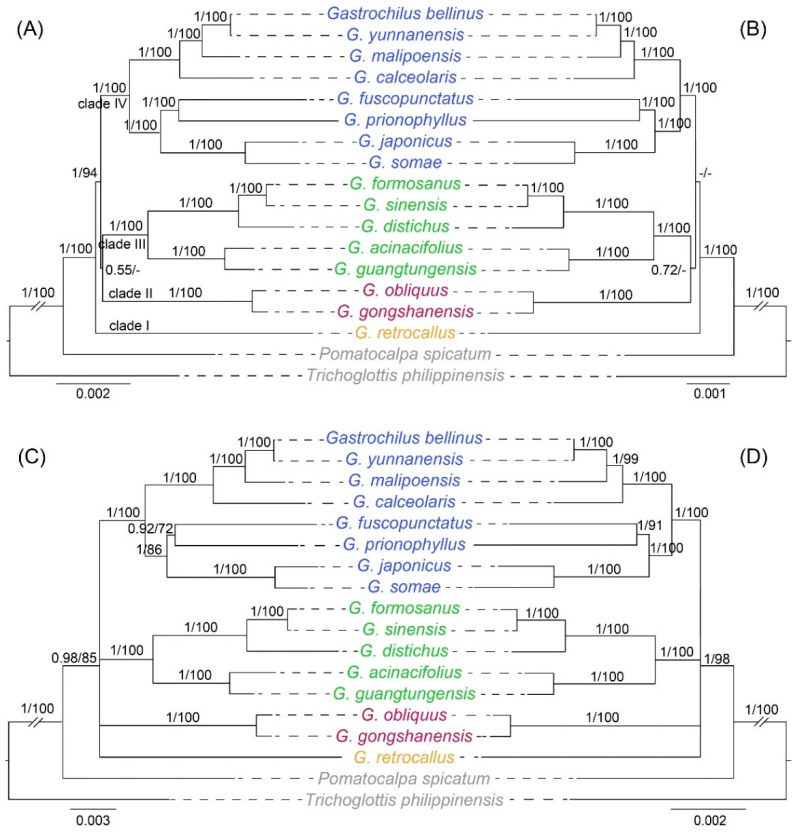
Comparisons of phylogenetic tree topologies for four datasets based on BI and ML analyses in *Gastrochilus* species. (**A**) Whole plastomes (excluding IRa). (**B**) Sixty-eight CDSs. (**C**) Top ten hypervariable regions of whole plastomes. (**D**) Top ten hypervariable regions of 68 CDSs. Numbers above the branches are Bayesian posterior probabilities and ML bootstrap values, respectively. A dash (-) indicates that the supporting values are less than 50%.

## Data Availability

All data are provided within this manuscript and Appendix A.

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
