# Peer review of "Plastome Evolution, Phylogenomics, and DNA Barcoding Investigation of Gastrochilus (Aeridinae, Orchidaceae), with a Focus on the Systematic Position of Haraella retrocalla"

_ijms, 2024, doi:10.3390/ijms25158500_

Round 1

Reviewer 1 Report

Comments and Suggestions for Authors

The reviewed paper investigates the evolution of playtime, phylogenomics, and DNA barcoding of the Asian epiphytic orchid genus Gastrochilus. The authors have recovered a well-resolved phylogeny and provided various molecular resources to further include species and molecular markers. Overall, the paper is easy to read, interesting, and the analyses are well done and sound. The paper is in good shape for publication after addressing some minor comments:

Introduction

- Lines 38-39: Do you mean the genus is monophyletic?

- Lines 39-41: If the relationships were not consistent, the authors must explain and fully describe these inconsistencies.

- Line 47: Do you mean because this species lacks...?

- Line 59: Please check the English.

- Line 62: This sounds like a random example, or is this group closely related to Gastrochilus?

- Lines 65-66: What do you mean in this sentence?

Discussion

- Line 213: Non-significant? Did authors perform statistical tests? Please provide p-values. The same comment applies to line 222. If it is an observation, please rephrase.

- Line 227: Say only "subclade." There is no need to say "monophyletic" as a clade is always monophyletic.

- Line 288: The improvement of molecular data? Please elaborate.

The authors did not fully discuss the repeats section or emphasize the significance of that analysis, focusing less on the molecular resources generated and more on the evolutionary implications. Please also further discuss Figures 2-3 and 5 in terms of evolution and comparative biology.

Comments on the Quality of English Language

Just need some minor reviews, I have commented on the lines that where not well written

Reviewer 2 Report

Comments and Suggestions for Authors

Revision of the manuscript “ Plastome evolution, phylogenomics and DNA barcoding investigation of Gastrochilus (Aeridinae, Orchidaceae), with a focus on systematic position of Haraella retrocalla” by Peng Zhou, Wan-Shun Lei,Ying-Kang Shi, Yi-Zhen Liu, Yan Luo, Ji-Hong Li and Xiao-Guo Xiang.

Review report

This paper deals with  phylogenetic relationships  of Gastrochilus  genus that count more than 70 species growing un  in tropical and subtropical habitat in Asia, comparing new (5) and published plastomes of 16 species using molecular markers and secondly making a focus  on the systematic position of the species Haraella retrocalla

I think this manuscript is interesting in its arguments because it results add new information on five Gastrochilus plastomes  

However there are some points that should be improved, in particular:

 First of all, you need to add the patronymic of the species the first time they are mentioned in the text and also made a reference to https://powo.science.kew.org/.

 Line 24-25: in the abstract it is not necessary to write  the possible application of the results obtained with this research. I suggest to delete.

 Figure 1 and Figure 2:  in the dendrograms you have to report the label of the 16 different species otherwise it is impossible to interpret the figures.

Line 259: regarding the study on the systematic position of H. retrocalla, I found one criticism. Even if the approach to identify the morphological differences between genera using statistical analysis such as principal component analysis (PCA) is correct, in this study, in my opinion it is useless given the macro differences (leaves, flowers, global morphology) which are visibly evident appreciable. I suggest moving the PCA as supplementary material and explaining in the text of the manuscript the main characters that differentiate the two genus which can also be appreciated in the photos.

 294: What is in your opinion a legal name? do the authors refer to the patronymic? Please to explain
